# Viral suppression and associated factors after enhanced adherence counseling among people living with HIV with unsuppressed viral loads at tertiary and first-level health facilities in Zambia: A retrospective cohort study

Chitalu Chanda[1]*, Webster C. Chewe[2], Benson M. Hamooya[3], Lukundo Siame[3,4], Matenge Mutalange[3], Aliness Dombola[5], Nyuma Mbewe[6], Chisha Sinyangwe[1], Melvin Mwansa[7], Duncan Chanda[1]

1 Department of Internal Medicine, University Teaching Hospital, Lusaka, Zambia, 2 Medical Department, AIDS Healthcare Foundation, Lusaka, Zambia, 3 School of Medicine and Health Sciences, Mulungushi University, Livingstone, Zambia, 4 Department of Internal Medicine, Livingstone University Teaching Hospital, Livingstone, Zambia, 5 Department of Public Health, National Health Research and Training Institute, Ndola, Zambia, 6 Department of Emergency Response, Zambia National Public Health Institute, Lusaka, Zambia, 7 Department of Public Health, University of Lusaka, Lusaka, Zambia

* chtalu@gmail.com

## Abstract

People living with HIV (PLHIV) who do not achieve viral suppression on antiretroviral therapy contribute to HIV transmission. Poor adherence is a major factor associated with high viral load (VL). Enhanced adherence counseling (EAC) is a targeted intervention to improve adherence and achieve viral suppression, but data on post-EAC outcomes in Zambia remain limited. This study assessed viral suppression and associated factors among PLHIV with unsuppressed VL after completion of EAC at University Teaching Hospital and Kanyama First-Level Hospital. This retrospective cohort study analyzed VL register data from 1st January 2021–31st December 2023. Baseline demographic, clinical, and laboratory data were collected, with follow-up VL measurements at three and 12 months post-EAC. The primary outcome was viral suppression at three months, defined as a VL < 200 copies/mL. Poisson regression with robust standard errors identified factors associated with suppression. Among 386 participants (median age 39 years, IQR: 31–47), 52.9% were female. The baseline VL was 21,600 copies/mL (IQR: 3,692–106,000). At three months post-EAC, 85% (330/386) achieved viral suppression, with 95.8% (316/330) maintaining suppression at 12 months. Viral rebound occurred in 4.2% (14/330). EAC delivered through both telephone and in-person methods increased suppression likelihood by 15% compared to those who received EAC in-person (physical) alone. Prior enrollment in six-month multi-month dispensing (MMD) was associated with a 23% increased likelihood of suppression compared to those who had never received MMD. Participants

**Data availability statement:** All data used in this study are publicly available in the manuscript and/or Supporting information files.

**Funding:** The author(s) received no specific funding for this work.

**Competing interests:** The authors have declared that no competing interests exist.

on tenofovir/lamivudine/dolutegravir were 29% more likely to suppress compared to those on zidovudine/lamivudine/dolutegravir. EAC modestly improves and sustains viral suppression among PLHIV with high viral loads. In-person and telephone-based EAC improved viral suppression by 15% compared to in-person alone. Other key factors influencing suppression were community-based delivery and prior six-month MMD. Findings highlight opportunities to integrate technology-enhanced adherence support and differentiated service delivery models to optimize HIV care outcomes.

## Introduction

Achieving high antiretroviral therapy (ART) coverage and viral suppression among people living with HIV (PLHIV) is critical to reducing HIV transmission, incidence, and related mortality [1]. Despite the expansion of ART programs, approximately 10% of PLHIV globally and 11% in sub-Saharan Africa remain with unsuppressed viral loads (VL), contributing to treatment failure [2,3]. Factors associated with poor viral suppression include male gender, younger age, stigma, substance use, low CD4 counts, and co-infections such as tuberculosis [2–4].

Recipients of Care (ROC) with unsuppressed VL, defined as HIV VL ≥ 1000 copies/mL after at least six months on ART, often face challenges related to adherence [5]. Enhanced Adherence Counseling (EAC) is a structured, patient-centered intervention designed to address adherence barriers and improve viral suppression. Delivered by trained healthcare providers at both facility and community levels, EAC has been shown to be effective, with observational studies reporting suppression rates between 61% and 73.8% in countries such as Zambia, Nigeria, and Ethiopia [6–8]. However, despite EAC's effectiveness, fewer than 10% of ROC fail to achieve viral suppression post-intervention [7,8]. Factors such as female gender, longer ART duration, second-line therapy, and baseline VL significantly influence EAC outcomes [9,10]. Additionally, innovations such as technology-enhanced EAC delivery and the use of dolutegravir (DTG)-based regimens have been linked to improved suppression rates [11,12].

In Zambia, the Ministry of Health has integrated EAC into routine HIV care to manage high VL (HVL) cases [13]. However, limited data exists on EAC outcomes and the factors influencing viral suppression across different healthcare settings. Existing studies often overlook key aspects, including sustained viral suppression beyond the EAC period, the impact of baseline VL on outcomes, and the role of facility-level factors in achieving suppression. Furthermore, viral suppression is often defined as VL < 1,000 copies/mL, despite growing emphasis on achieving suppression below 200 copies/mL to effectively reduce HIV transmission [8,9,14].

Therefore, this study retrospectively analyzes the incidence of viral suppression below 200 copies/mL and associated factors among PLHIV with unsuppressed VL at a tertiary and a first-level healthcare facility in Lusaka, Zambia. The findings aim to provide insights into the effectiveness of EAC and inform strategies for optimizing HIV care delivery.

## Methodology

### Ethics statement

The study protocol was reviewed and approved by the University of Lusaka Research Ethics Board (UNILUSREC) (FWA00033228–203(08)/(08)/{2024}) and the National Health Research Authority (NHRA) (1775/10/12/2024). A waiver of informed consent was granted due to the retrospective nature of the study. To ensure confidentiality and protect participant privacy, all personal identifiers, including names, dates of birth, and other identifiable information, were removed from the dataset.. To ensure transparent reporting, we adhered to the Strengthening the Reporting of Observational Studies in Epidemiology (STROBE) guidelines (see S1 Checklist).

### Study design and setting

This retrospective cohort study analyzed data from PLHIV with HVL (VL ≥ 1,000 copies/mL) who were enrolled in EAC at University Teaching Hospital (tertiary level) and Kanyama First-Level Hospital (primary level) in Lusaka, Zambia. The study covered a period from January 1, 2021, to December 31, 2023, utilizing routine program data recorded in the unsuppressed VL or HVL registers at baseline, three months, and 12 months post-EAC. Enhanced Adherence Counselling (EAC) is a structured, client-centered intervention for people living with HIV who have unsuppressed viral loads (≥1,000 copies/mL). National ART guidelines recommend at least three individual sessions, usually one month apart, focused on identifying and addressing adherence barriers, reinforcing optimal ART use, and developing personalized adherence plans. Sessions are delivered by trained health workers or lay counsellors using standard tools, with repeat viral load testing after completion to assess response. The initial EAC visit is normally offered at the health facility, while follow up visit may be offered in the community. Some follow up EAC services are conducted via telemedicine which involves phone calls conversation with clients unable to attend facility visits.

### Eligibility criteria

The study included participants aged 16 years or older who had been diagnosed with HIV and on ART for at least six months. Eligible participants had a recorded high viral load (≥1,000 copies/mL) at baseline and were enrolled in EAC. Individuals were excluded if they lacked documented viral load measurements at three and 12 months post-EAC.

### Sample size estimation

The combined treatment cohort across both facilities included over 14,786 ROC. The estimated number of clients with HVL during the study period served as the population base. Assuming a viral suppression rate of 61% and a non-completion rate of 8–10% among those enrolled in EAC, using proportions at 95% confidence interval with a 5% margin of error, a sample of 386 participants was recruited [8]. A convenience sampling approach was used to include all eligible clients, given the limited recording of HVL cases.

### Data collection

Between 31st December 2024 and 08th January, 2025, data for the period 1st January, 2021–31st December, 2023, was extracted from medical records and HIV registers using a structured form designed in Research Electronic Data Capture (REDCap) [15]. Collected variables included socio-demographics (age, sex, residential address, marital status, partner status, and occupation), ART-related information (date of HIV diagnosis, ART initiation date, duration on ART, EAC enrollment date, mode of EAC delivery, adherence history, and ART regimen changes), and HIV viral load measurements at baseline, three months, and 12 months post-EAC. Additionally, data on advanced HIV disease screening, including cryptococcal antigen testing and tuberculosis history, were obtained.

## Outcome of interest

The primary outcome of interest was viral suppression, defined as an HIV viral load of less than 200 copies/mL at three months, corresponding to the completion of EAC [13,15,16]. Secondary outcomes included persistent HVL, defined as an HIV VL of ≥1,000 copies/mL after EAC completion, and HIV viral rebound, defined as an HIV VL of ≥1,000 copies/mL after initially achieving suppression (<200 copies/mL) following EAC [13,16,17].

## Data analysis

Data were cleaned in Excel and analyzed using STATA version 15 [18]. Baseline characteristics were summarized using medians and interquartile ranges (IQRs) for continuous variables, as these were determined to be skewed using the Shapiro-Wilk test. Categorical variables were presented as proportions and frequencies. Associations between categorical variables were assessed using the chi-square test or Fisher's exact test, as appropriate. Changes in viral load outcomes at three and 12 months were analyzed using the Wilcoxon signed-rank test. To identify factors associated with viral suppression at three months post-EAC, Poisson regression with robust standard errors was used. Variables included in the multivariable model were selected based on previous literature and those with a p-value <0.05 in univariate analysis. Statistical significance was set at $p < 0.05$.

## Results

### Study participants' characteristics

The study included 386 participants from both a primary and tertiary health facility, followed between January 1, 2021, and December 31, 2023 as shown in Fig 1. The median age was 39 years (IQR: 31–47), with 52.9% (204/386) being female. Most participants (75.4%, 291/385) resided within the facility catchment area.

Before EAC enrollment, 22% (85/386) were receiving six-month multi-month dispensing (MMD). The majority (69.8%, 266/381) were on a Tenofovir Disoproxil Fumarate/Lamivudine/Dolutegravir (TLD) regimen. The median CD4 count was 295 cells/μL (IQR: 74–350). Nearly half (48.7%, 188/386) initiated EAC within one week of receiving high viral load results, with a median viral load of 21,600 copies/mL (IQR: 3,692–106,000) as shown in Table 1.

### HIV viral suppression after EAC

At three months post-EAC, 85% (330/386) of participants achieved viral suppression. By the 12-month follow-up, 95.8% (316/330) of those initially suppressed maintained suppression. Among participants who remained unsuppressed after three months, 57.1% (32/56) later achieved suppression at 12 months, while 42.9% (24/56) remained unsuppressed. Viral rebound occurred in 4.2% (14/330) of those who had initially suppressed following EAC, as shown in Table 2.

### Changes in viral load from baseline to 3 months and 12 months

At baseline, the median viral load was 21,600 copies/mL (IQR: 3,692–106,000). After 3 months of follow-up, there was a substantial decline to a median of 20 copies/mL (IQR: 0–61), and by 12 months the viral load further decreased to 0 copies/mL (IQR: 0–21), see Fig 2.

### Factors associated with viral suppression

Participants who received EAC through a combination of telephone and in-person methods were 15% more likely to achieve viral suppression (aIRR: 1.15, 95% CI: 1.01–1.31; p = 0.002). Conversely, those receiving a mix of community- and facility-based EAC were less likely to attain suppression compared to those who received community-based EAC

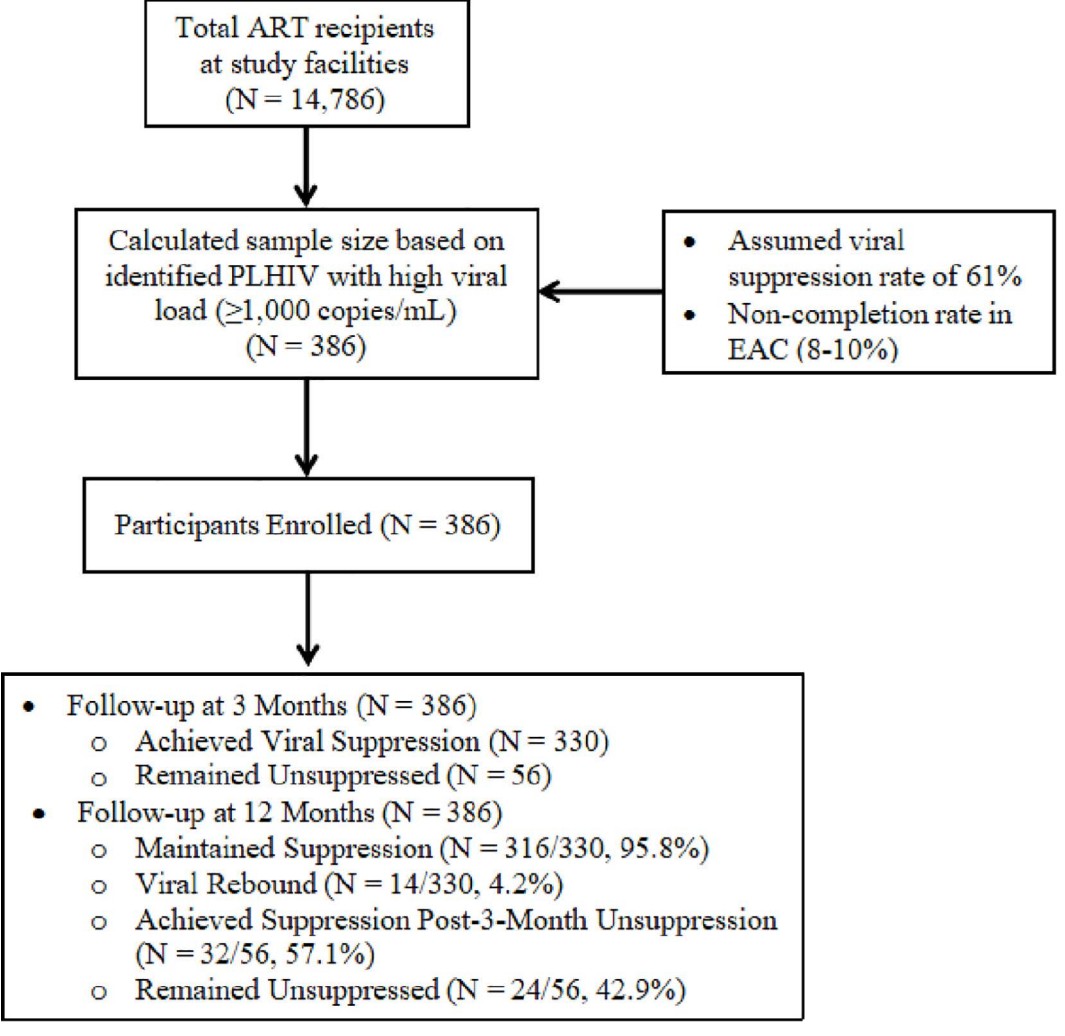

**Fig 1. Flowchart for the recruitment process of study participants.**

alone (aIRR: 0.93, 95% CI: 0.81–1.06; p = 0.42). Participants on AZT/3TC/DTG regimen were significantly less likely to achieve viral suppression compared to those on a TLD regimen (aIRR: 0.71, 95% CI: 0.51–0.91; p = 0.007). Additionally, prior enrollment in six-month MMD was associated with a 23% increased likelihood of achieving suppression (aIRR: 1.23, 95% CI: 1.06–1.42; p = 0.01), as shown in Table 3.

## Discussion

This study demonstrated that HIV viral suppression was achieved in 85.5% of PLHIV with high viral loads after three months of EAC. Furthermore, 95.8% of those who attained suppression sustained it at the 12-month follow-up. These viral suppression rates are higher than those reported in Zimbabwe (74.2%), Ethiopia (66.4%), Uganda (66.4%), and Nigeria (66.6%) [6,9,11,12]. The improved outcomes observed in this study may be attributed to the increased use of dolutegravir (DTG)-based regimens, a mixed adherence counseling model (telephone and physical), fewer adolescents in the cohort, and more frequent counseling sessions during the three-month EAC period. Achieving viral suppression below 200

**Table 1. Basic demographic and clinical characteristics.**

| Variable | Number | Median (IQR) | Frequency (%) |
|---|---|---|---|
| Age | 386 | 39 (31, 47) | |
| Sex | 386 | | |
| Male | | | 182 (47.2) |
| Female | | | 204 (52.9) |
| Education | 385 | | |
| Primary | | | 14 (3.6) |
| Secondary | | | 59 (15.3) |
| Unknown | | | 312 (81.0) |
| Level of facility | 385 | | |
| Tertiary | | | 94 (24.4) |
| Primary | | | 291 (75.6) |
| Residence to facility | 386 | | |
| Within catchment | | | 291 (75.4) |
| Outside catchment | | | 2 (0.52) |
| Unknown | | | 93 (24.1) |
| Partner HIV status | 384 | | |
| Positive | | | 40 (10.4) |
| Negative | | | 9 (2.3) |
| Unknown | | | 335 (87.2) |
| HIV indexing Testing | 385 | | |
| No | | | 2 (0.5) |
| Yes | | | 299 (77.7) |
| Unknown | | | 84 (21.8) |
| Mode of EAC given | 386 | | |
| Telephone | | | 0 (0.0) |
| Physical | | | 93 (24.1) |
| Both | | | 293 (75.9) |
| Place of EAC Given | 386 | | |
| Community | | | 2 (0.52) |
| Facility | | | 271 (70.2) |
| Combined | | | 113 (29.7) |
| Patient on 6-month MMD | 386 | | |
| Yes | | | 85 (22.0) |
| No | | | 301 (78.0) |
| ART regimen | 381 | | |
| TLD | | | 266 (69.8) |
| TAFeD | | | 55 (14.4) |
| AZT/3TC/DTG | | | 37 (9.7) |
| TLE | | | 23 (6.0) |
| Baseline CD4 at EAC enrolment, cells/mm3 | 22 | 295 (174,350) | |
| Baseline weight, Kg | 31 | 59 (55,65) | |
| Prior EAC session | | | |
| Yes | | | 97 (25.2) |
| No | | | 288 (74.8) |
| Time to Commencement of EAC | 386 | | |
| Within 1 week | | | 188 (48.7) |
| 1 week to 4 weeks | | | 68 (17.6) |
| After 4 weeks | | | 130 (33.7) |

*(Continued)*

**Table 1.** (Continued)

| Variable | Number | Median (IQR) | Frequency (%) |
|---|---|---|---|
| Number of EAC session | 386 | 6 (4,6) | |
| Viral load before EAC, copies/mL | 386 | 21 600 (3692,106000) | |
| Viral load after EAC (3 Months), copies/mL | 386 | 20 (0, 61) | |
| Presence of Opportunistic Tuberculosis | 386 | | |
| Yes | | | 4 (1.0) |
| No | | | 382 (99.0) |
| Duration on ART, months | | 94.99 (66.1, 157.4) | |

Abbreviation: ART: antiretroviral therapy; AZT/3TC/DTG: Zidovudine/Lamivudine/Dolutegravir; EAC: enhanced advanced counseling; IQR: interquartile range; MMD: multiple monthly dispersion; TAFeD: Tenofovir Alafenamide, Emtricitabine, and Dolutegravir; TLD: Tenofovir, Lamivudine, and Dolutegravir; TLE: Tenofovir, Lamivudine, and Efavirenz.

Footnote: HIV index testing involves provision of HIV testing to all partners and family members of the recipient of care with high viral load.

**Table 2.** HIV viral suppression 3 months after EAC and 12 Months post EAC.

| At three (3) months, n (%) | | At 12 months, n (%) | |
|---|---|---|---|
| | | **Suppressed** | **Unsuppressed** |
| Suppressed | 330 (85.5) | 316 (95.8) | 14 (4.2) |
| Unsuppressed | 56 (14.5) | 32 (57.1) | 24 (42.9) |

**Note:** Proportions in parentheses are calculated as *numerator ÷ denominator × 100*.

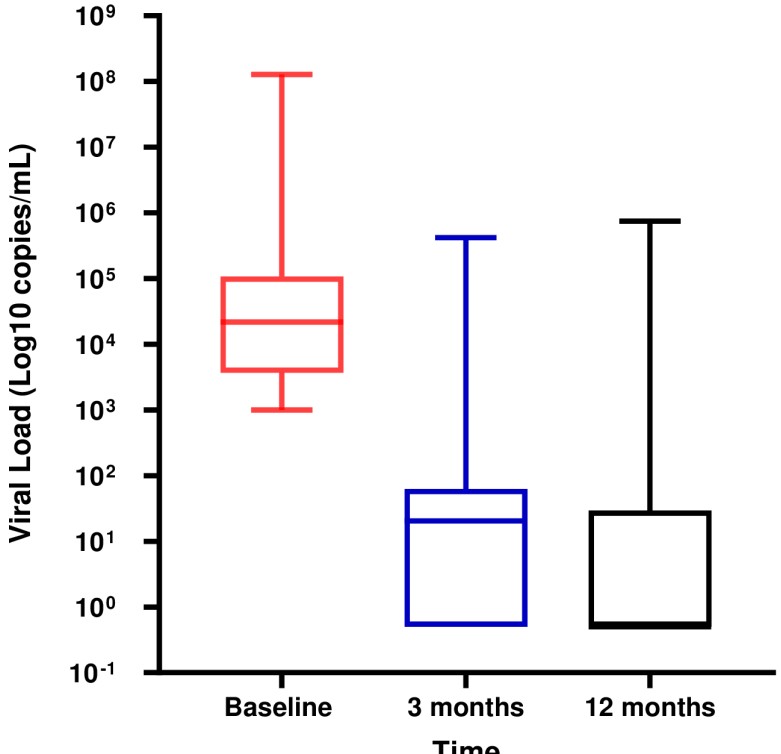

**Fig 2. Changes in viral load from baseline to 3 months and 12 months.** *Paired Wilcoxon signed-rank test* p < 0.0001.

**Table 3. Regression analysis of factors associated with HIV viral suppression at 3 months after EAC.**

| Variables | cIRR (95%CI) | P-value | aIRR (95%CI) | p-value |
|---|---|---|---|---|
| Age | 1.00 (0.99, 1.01) | 0.222 | 1.00 (0.99, 1.01) | 0.339 |
| Sex | | | | |
| Male | ref | | ref | |
| Female | 1.06 (0.97, 1.15) | 0.189 | 1.04 (0.96, 1.13) | 0.312 |
| Level of facility | | | | |
| Tertiary | ref | | ref | |
| Primary | 1.11 (0.99, 1.25) | 0.065 | 0.86 (0.53, 1.38) | 0.520 |
| Educational level | | | | |
| Primary | ref | | ref | |
| Secondary | 1.42 (0.89, 2.28) | 0.141 | 1.36 (0.92, 2.01) | 0.113 |
| Unknown | 1.53 (0.97, 2.42) | 0.067 | 1.33 (0.88, 2.04) | 0.177 |
| Mode of EAC | | | | |
| Physical (Within the clinic) | ref | | ref | |
| Telephone and physical | 1.12 (0.99, 1.25) | 0.058 | 1.15 (1.01, 1.31) | **0.002** |
| Place of EAC | | | | |
| Community | ref | | ref | |
| Facility EAC | 0.84 (0.79, 0.88) | **0.001** | 0.90 (0.80, 1.00) | 0.070 |
| Combined | 0.89 (0.84, 0.95) | **0.001** | 0.93 (0.81, 1.06) | **0.042** |
| Duration on ART | 0.99 (0.99, 1.00) | 0.461 | 0.99 (0.99, 1.00) | 0.309 |
| ART regimen | | | | |
| TLD | ref | | ref | |
| TAFeD | 1.03 (0.94, 1.14) | 0.497 | 1.00 (0.89, 1.13) | 0.975 |
| AZT/3TC/DTG | 0.68 (0.52, 0.89) | **0.004** | 0.71 (0.56, 0.91) | **0.007** |
| TLE | 1.04 (0.91, 1.19) | 0.586 | 1.18 (0.97, 1.43) | 0.091 |
| Patient on 6 MMD | | | | |
| No | ref | | ref | |
| Yes | 1.17 (1.03, 1.33) | **0.015** | 1.23 (1.06, 1.42) | **0.010** |
| Number of EAC session | 1.01 (0.97, 1.06) | 0.525 | 0.95 (0.88, 1.03) | 0.216 |
| Time to Commencement of EAC | | | | |
| Within 1 week | ref | | ref | |
| 1 week to 4 weeks | 1.11 (1.00, 1.22) | **0.047** | 1.06 (0.95, 1.18) | 0.333 |
| After 4 weeks | 1.05 (0.96, 1.15) | 0.270 | 0.99 (0.89, 1.10) | 0.860 |
| Prior EAC session | | | | |
| No | ref | | ref | |
| Yes | 0.87 (0.78, 0.98) | **0.022** | 0.83 (0.58, 1.22) | 0.353 |

Abbreviation: aIRR: adjusted incidence rate ratio; ART: antiretroviral therapy; AZT/3TC/DTG: Zidovudine/Lamivudine/Dolutegravir; cIRR: crude incidence rate ratio; EAC: enhanced advanced counseling; MMD: multiple monthly dispersion; TAFeD: Tenofovir alafenamide, Emtricitabine, and Dolutegravir; TLD: Tenofovir, Lamivudine, and Dolutegravir; TLE: Tenofovir, Lamivudine, and Efavirenz.

copies/mL is critical, as it eliminates the risk of HIV transmission, aligning with the "undetectable = untransmittable (U=U)" principle [19].

Despite the positive outcomes, 4.2% of participants who initially achieved viral suppression experienced viral rebound at the 12-month follow-up, a rate comparable to the 21.0% found in Ghana [20]. These viral increases could be viral blips. Further, other factors could be contributing to viral rebound such include poor adherence, treatment interruptions, and co-infections such as tuberculosis [21,22]. These findings emphasize the importance of continued adherence counseling, even for individuals who have attained viral suppression.

The study also found that a combination of telephone and physical EAC significantly improved the likelihood of viral suppression, supporting findings from previous studies that demonstrated the effectiveness of telephone-supported EAC in facilitating engagement and completion of counseling sessions [11,23]. This highlights the potential for technology-enhanced interventions to improve adherence counseling outcomes.

Participants on a single-pill TLD regimen were more likely to achieve viral suppression compared to those on AZT/3TC/DTG. This aligns with findings from Ethiopia, where single-pill formulations were associated with better adherence and viral suppression outcomes due to reduced pill burden [24]. However, a study in Uganda suggested that the presence of DTG alone was sufficient to enhance suppression, regardless of the formulation [12]. This discrepancy highlights the need for further research to clarify the influence of ART regimens on viral suppression outcomes.

Additionally, community-based EAC delivery was associated with higher viral suppression rates compared to facility-based EAC, consistent with evidence supporting differentiated service delivery (DSD) models for improved retention and clinical outcomes [25]. These findings suggest that expanding DSD models, particularly for clients with HVL, could enhance accessibility and effectiveness.

Participants previously enrolled in six-month MMD were also more likely to achieve viral suppression following EAC. This association may reflect better adherence practices among these clients, indicating the potential benefits of continuing shorter-duration MMD for individuals with HVL to support sustained adherence.

## Study limitations and strengths

This study had some limitations. Due to the retrospective study design, certain variables, such as occupation, which could provide further insights into adherence patterns, were not available. Additionally, genotype sequencing was not conducted to determine HIV drug resistance, which could influence viral suppression outcomes. A key strength of the study was its long-term follow-up period, which allowed for analyzing sustained viral suppression beyond the initial EAC phase. This provides valuable insights into long-term treatment success and potential gaps in adherence support.

## Conclusions

The findings underscore the effectiveness of EAC in achieving viral suppression among PLHIV with high viral loads. Key factors influencing viral suppression include the combined use of telephone and physical counseling, prior enrollment in six-month MMD, and community-based EAC delivery. These results highlight the importance of integrating innovative and differentiated care models to optimize treatment outcomes. Further research should explore the use of technology and community-based interventions in the provision of adherence support for PLHIV with unsuppressed viral loads.

## Recommendations

1. Strengthen and expand community-based EAC delivery to improve accessibility and effectiveness.

2. Utilize both telephone and physical counseling methods to enhance engagement and adherence.

3. Continue adherence counseling even after viral suppression is achieved to prevent viral rebound.

4. Future studies should assess the impact of EAC on HIV transmission dynamics and long-term treatment outcomes.

## Supporting information

**S1 Checklist. STROBE statement—Checklist of items that should be included in reports of cross-sectional studies.**
(DOCX)

**S1 Data. Dataset for clients with high viral load**
(XLS)

## Acknowledgments

We thank the management of University Teaching Hospital (UTH) and Kanyama First-Level Hospital for granting access to hospital data.

## Author contributions

**Conceptualization:** Chitalu Chanda, Webster C. Chewe.

**Formal analysis:** Chitalu Chanda, Webster C. Chewe, Benson M. Hamooya, Lukundo Siame, Matenge Mutalange.

**Methodology:** Chitalu Chanda, Webster C. Chewe, Benson M. Hamooya.

**Supervision:** Melvin Mwansa, Duncan Chanda.

**Writing – original draft:** Chitalu Chanda, Webster C. Chewe, Matenge Mutalange, Aliness Dombola, Nyuma Mbewe, Chisha Sinyangwe, Melvin Mwansa, Duncan Chanda.

**Writing – review & editing:** Chitalu Chanda, Webster C. Chewe, Benson M. Hamooya, Lukundo Siame, Matenge Mutalange, Aliness Dombola, Nyuma Mbewe, Chisha Sinyangwe, Melvin Mwansa, Duncan Chanda.

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
