## [Decision Letter · Decision Letter 0]

19 Jun 2025

PGPH-D-25-00977

Viral suppression and associated factors after enhanced adherence counseling among people living with HIV with unsuppressed viral loads at tertiary and first-level health facilities in Zambia: A retrospective cohort study

Dear Dr. chanda,

Thank you for submitting your manuscript to PLOS Global Public Health. After careful consideration, we feel that it has merit but does not fully meet PLOS Global Public Health’s publication criteria as it currently stands. Therefore, we invite you to submit a revised version of the manuscript that addresses the points raised during the review process.

We look forward to receiving your revised manuscript.

Kind regards,

Jianhong Zhou

Staff Editor

Journal Requirements:

Additional Editor Comments (if provided):

Reviewers' comments:

Reviewer's Responses to Questions

**Comments to the Author**

1. Does this manuscript meet PLOS Global Public Health’s publication criteria?

Reviewer #1: Yes

Reviewer #2: Partly

2. Has the statistical analysis been performed appropriately and rigorously?

Reviewer #1: No

Reviewer #2: Yes

3. Have the authors made all data underlying the findings in their manuscript fully available (please refer to the Data Availability Statement at the start of the manuscript PDF file)?

Reviewer #1: No

Reviewer #2: Yes

4. Is the manuscript presented in an intelligible fashion and written in standard English?

Reviewer #1: Yes

Reviewer #2: Yes

Reviewer #1: Thank you for the opportunity to review this manuscript. The manuscript was well written and describes the effectiveness of enhanced adherence counselling among individuals with high HIV viral loads at three months and 12 months post initiation of EAC. The manuscript also describes factors that were associated with achieving viral suppression in the same population of PLHIV. While the paper reads fairly easily there are a number of methodological flaws and inconsistencies in the reporting that the authors need to address. The main one is that it’s not clear what the outcome for the analysis was ie to compare the % virally suppressed post EAC among those who received EAC vs those who did not (implied in the abstract) OR to compare the effectiveness between those who received EAC via telephone only vs those who received EAC in-person and via telephone (as implied and presented in the results). I have listed these and other concerns of these in the comments below

• Abstract

o Introduction- the authors state the objectives as to describe viral suppression and associated factors without mentioning that this was post EAC. This makes the preceding discussion of EAC seem odd

o Results – in the results section the authors reported that EAC delivered both in-person and via the telephone increased the likelihood of viral suppression by 15%. They don’t state at what time point- 3m or 12 m or at either time point and they also don’t state what the comparison group was. Without stating explicitly was the comparison group was it was easy to assume it was vs those who did not receive EAC which didn’t make sense as all people enrolled were rolled at initiation of EAC

o Results – for the other factors – MMD and ART regimen, can the authors also state what the comparison/ reference groups were

o The authors concluded that EAC effectively improved and sustained viral suppression again. This seems like overstating the effect. I would say that EAC only modestly improves VL suppression. 15% is not too impressive and there was the question of DTG roll-out in the same period. Also it should say in-person + telephone based improved viral load suppression by 15% compared to telephone alone as this is what the data showed

• Introduction

o In Lines 65 – 68 the authors wrote that EAC was shown to be effective in a achieving viral suppression by 61- 73.8% in the countries listed. Was this under trial conditions?

• Methodology

o In the section on study design and setting, may the authors say a bit more about these hospitals and the services they provided? What are their catchment areas and did they refer to one another? Why pick these two out of all eligible facilities in Lusaka?

o In the section on sample size calculations, were there any other assumptions made about the sample size. From the information provided I cannot replicate this sample size calculation. Were the viral suppression levels discussed post EAC? Also what precision level was assumed in the sample size calculations?

o Regarding Figure 1, are the numbers in Figure 1 what was found after data collection was completed or what was assumed at study design. If this is what was found, please move to the results. It will be good of the authors can also include a consort diagram in the results

o The outcome of interest was VL<200 copies/ml at 3m. This piece of information should be added to the abstract. Also why 200 copies/ ml?

• Results

o Line 150 should be areas since there were two facilities

o Line 153 - at what point was this CD4 count measured? Also in the setting and background, say something about CD4 count monitoring or measurement in the context of PLHIV on ART and having a high viral load or starting EAC

o Table 1: how does the % of males/ females in this population compare with that of PLHIV in care and on ART?

o Table 1: regarding mode of EAC, shouldn’t this be categorized as physical only vs physical and telephone. Why add telephone only when there was no- one who received it

o Table 1: regarding baseline CD4 count at enrolment, please see earlier comment. Is this baseline as in the start of ART/ entry into care? What is the policy around CD4 count monitoring and EAC / loss of viral suppression?

o Table 1: what is meant by HIV indexing

o Table 1: duration on ART: what is this, median with IQR. If so say so

o Table 2: add the numbers of patients whose viral loads were measured to get these medians and IQR. Also indicate what that the test was the Wilcoxon sign rank test p-value

o Table 4: This table is mixed up. For some variables the authors present column totals and for some they present row totals. Please present column totals for all variables. Alternatively remove table 4 and incorporate the information provided in it in what is table 5 as n/N for each category of variables

o Table 4: I don’t understand the variable place of EAC. Is it for those who got in-person. How was the community based EAC provided and by who? Maybe add to the settings section

o Line 183: compared to what

• Discussion

o The so-what of the paper doesn’t come out clearly. As mentioned earlier, telephone + in-person achieved a modest improvement in VL suppression. Given the most effect, was there an understanding of costs involved and if the costs of telephone vs in-person are balanced by the risk of unsuppressed VLs. Was there any data form literature on alternatives to the telephone eg whatsapp chat bots etc

Reviewer #2: This manuscript explored factors associated with viral suppression for people living with HIV (PLHIV) post-intervention (Enhanced Adherence Counseling: EAC). Viral suppression was assessed 3- and 12-months post EAC (e.g., mode, place administered). Study findings suggests viral suppression was achieved at 3-months (85%) and sustained at 12-months (96%). EAC delivered through a combination of telephone and in-person sessions increased the likelihood of suppression. It was also noted that enrollment in the 6-month multi-month dispensing (MMD) taking tenofovir/lamivudine/dolutegravir were more likely to achieve suppression. This manuscript is well-written and covers the important topic of viral suppression and targeted intervention to improve and sustain suppression. The study findings can be important when developing interventions to address viral suppression. However, the limited study methodology makes it difficult to determine just how impactful the manuscript will be to the related literature. Given the importance of the intervention (EAC) in relation to the study outcomes, more information needs to be included about EAC as well as other factors believed to impact viral suppression.

Specifically:

1. There is very little information here regarding EAC, specifically: 1) What does the three modes of EAC look like in practice?; 2) What is contained in the EAC sessions?; 3) “Places EAC Given” refers to what exactly?; 4) What is meant by “prior EAC?”; and 5) When referring to time to EAC, are there other interventions during this time or just a waitlist?

2. The manuscript briefly mentions 6-month MMD, were other options available?

3. What are the differences between primary and tertiary facilities (specifics can vary for area to area).

4. It is noted that the sample is made up of adults age 31 and old (median 39), and given that younger PLHIV (ages 16 - 29) are a population most impacted by the lack of viral suppression, it may be important to specifically note how this study can possibly impact this subgroup.

5. Occupation is very important with regard to adherence patterns, and while data was not available, suggestions for future directions is important.

**Do you want your identity to be public for this peer review?** For information about this choice, including consent withdrawal, please see our Privacy Policy

Reviewer #1: No

Reviewer #2: No

---

## [Decision Letter · Decision Letter 1]

27 Aug 2025

PGPH-D-25-00977R1

Viral suppression and associated factors after enhanced adherence counseling among people living with HIV with unsuppressed viral loads at tertiary and first-level health facilities in Zambia: A retrospective cohort study

Dear Dr. chanda,

Thank you for submitting your manuscript to PLOS Global Public Health. After careful consideration, we feel that it has merit but does not fully meet PLOS Global Public Health’s publication criteria as it currently stands. Therefore, we invite you to submit a revised version of the manuscript that addresses the points raised during the review process.

We look forward to receiving your revised manuscript.

Kind regards,

Orvalho Augusto, MD, MPH, PhD

Academic Editor

Journal Requirements:

Additional Editor Comments (if provided):

This is a well-written manuscript, and the authors did a good job responding to the questions raised by the reviewers. However, there are still issues to be resolved:

1. To enhance the quality of reporting of a cohort study, it is recommended to use the STROBE for the cohort study checklist (https://www.strobe-statement.org/checklists/). Please fill out the form and add it to the manuscript as supplementary materials. And please address the relevant items, if possible, in the text.

2. A comment: The adjusted analysis has the issue of using p-values to choose what to adjust for. This is a bad practice when the goal is to find potential factors. And if it is done, the p-value (to choose variables to adjust for) should not be so stringent. Nevertheless, it appears that age and sex were forced into the model (which is good!).

A question: And why is “age” included as a linear covariate? This means its “effect” is expected to be the same for the range of ages. Is that plausible?

3. We need a good paragraph, either in the introduction or in the methods, explaining what EAC consists of in Zambia. One reviewer asked for this already.

4. Please Cite REDCap and Stata.

5. Table 1 - Should be only for baseline, i.e, characteristics at enrolment to EAC. Please revise.

- What is “HIV indexing done”?

6. Table 2 - better be presented as a figure with three boxplots (and the Y-axis in logarithmic scale, add 0.5 before the logarithm). Move table 2 to the annexes.

7. Table 3 - Add a note explaining how the proportions are calculated

8. Table 4:

- Add a column with prevalence of “Viral suppression after EAC”

- We do not need p-values in this table, especially with almost the same p-values on the unadjusted analysis in table 5.

- Add a footnote explaining how the proportions are computed

Reviewers' comments:

Reviewer's Responses to Questions

**Comments to the Author**

1. If the authors have adequately addressed your comments raised in a previous round of review and you feel that this manuscript is now acceptable for publication, you may indicate that here to bypass the “Comments to the Author” section, enter your conflict of interest statement in the “Confidential to Editor” section, and submit your "Accept" recommendation.

Reviewer #1: All comments have been addressed

2. Does this manuscript meet PLOS Global Public Health’s publication criteria ? Is the manuscript technically sound, and do the data support the conclusions? The manuscript must describe methodologically and ethically rigorous research with conclusions that are appropriately drawn based on the data presented.

Reviewer #1: Yes

3. Has the statistical analysis been performed appropriately and rigorously?

Reviewer #1: Yes

4. Have the authors made all data underlying the findings in their manuscript fully available (please refer to the Data Availability Statement at the start of the manuscript PDF file)?

Reviewer #1: No

5. Is the manuscript presented in an intelligible fashion and written in standard English?

Reviewer #1: Yes

6. Review Comments to the Author

Reviewer #1: Thank you for the opportunity to re-review this manuscript. The authors did a good job of addressing the comments I had. I am still not clear how the sample size was determined. I assumed viral suppression of 61% +/- 5% at 3 months, an alpha level of 5% and 80% power. This gives a sample size of 189 using the Score test method. If the sample size is inflated by 10%, it still doesnt give 386. Maybe the authors can calculate power for the realized sample size instead

7. PLOS authors have the option to publish the peer review history of their article (what does this mean? ). If published, this will include your full peer review and any attached files.

**Do you want your identity to be public for this peer review?** For information about this choice, including consent withdrawal, please see our Privacy Policy .

Reviewer #1: No

---

## [Decision Letter · Decision Letter 2]

19 Oct 2025

Viral suppression and associated factors after enhanced adherence counseling among people living with HIV with unsuppressed viral loads at tertiary and first-level health facilities in Zambia: A retrospective cohort study

PGPH-D-25-00977R2

Dear Dr chanda,

We are pleased to inform you that your manuscript 'Viral suppression and associated factors after enhanced adherence counseling among people living with HIV with unsuppressed viral loads at tertiary and first-level health facilities in Zambia: A retrospective cohort study' has been provisionally accepted for publication in PLOS Global Public Health.

Best regards,

Orvalho Augusto, MD, MPH, PhD

Academic Editor

Few comments:

1. Table 2 - Indicate which proportions are in rows and which are in columns.

2. The paired Wilcoxon signed-rank test is for paired measures of individuals. Please indicate which pairs are (eg: 3 months vs Baseline?)

3. In the abstract make sure all p-values have 3 decimal places

4. Many write STATA but the official way of writing is Stata because it isn't an acronym. Check the official documentation.

Reviewer Comments (if any, and for reference):

Reviewer's Responses to Questions

**Comments to the Author**

Reviewer #1: (No Response)

publication criteria?

Reviewer #1: Yes

3. Has the statistical analysis been performed appropriately and rigorously?

Reviewer #1: Yes

4. Have the authors made all data underlying the findings in their manuscript fully available (please refer to the Data Availability Statement at the start of the manuscript PDF file)?

Reviewer #1: No

5. Is the manuscript presented in an intelligible fashion and written in standard English?

Reviewer #1: Yes

Reviewer #1: Thank you for the opportunity to re-review this paper. I think the paper reads much better than previous versions and have no further comments. The authors did not address the comment on sample size calculation from the two previous reviews. They may not want to address it (in which case they should say so and why) and I don't think its a deal breaker for the paper. However I think its bad manners to ignore comments from reviewers

**Do you want your identity to be public for this peer review?** For information about this choice, including consent withdrawal, please see our Privacy Policy

Reviewer #1: No
